# Ancestry-informative markers for African Americans based on the Affymetrix Pan-African genotyping array

Xu Zhang[1,2], Wenbo Mu[3], Cong Liu[3] and Wei Zhang[1,3]

[1] The Affiliated Hospital of Medical School, Ningbo University, Ningbo, Zhejiang Province, China
[2] Section of Hematology/Oncology, Department of Medicine, University of Illinois, Chicago, IL, USA
[3] Department of Pediatrics, University of Illinois, Chicago, IL, USA

Corresponding author
Wei Zhang,
weizhang.chicago@gmail.com

## ABSTRACT

Genetic admixture has been utilized as a tool for identifying loci associated with complex traits and diseases in recently admixed populations such as African Americans. In particular, admixture mapping is an efficient approach to identifying genetic basis for those complex diseases with substantial racial or ethnic disparities. Though current advances in admixture mapping algorithms may utilize the entire panel of SNPs, providing ancestry-informative markers (AIMs) that can differentiate parental populations and estimate ancestry proportions in an admixed population may particularly benefit admixture mapping in studies of limited samples, help identify unsuitable individuals (e.g., through genotyping the most informative ancestry markers) before starting large genome-wide association studies (GWAS), or guide larger scale targeted deep re-sequencing for determining specific disease-causing variants. Defining panels of AIMs based on commercial, high-throughput genotyping platforms will facilitate the utilization of these platforms for simultaneous admixture mapping of complex traits and diseases, in addition to conventional GWAS. Here, we describe AIMs detected based on the Shannon Information Content (SIC) or $F_{st}$ for African Americans with genome-wide coverage that were selected from ∼2.3 million single nucleotide polymorphisms (SNPs) covered by the Affymetrix Axiom Pan-African array, a newly developed genotyping platform optimized for individuals of African ancestry.

Subjects Bioinformatics, Genetics
Keywords Admixture mapping, Single nucleotide polymorphism, Pan-African array, Ancestry-informative marker, African American

## INTRODUCTION

High throughput genotyping arrays have facilitated genome-wide association studies (GWAS) on complex traits (*Hindorff et al., 2009*) including risks for common, complex diseases and drug response. In contrast to a conventional GWAS in a homogeneous parental populations (e.g., Caucasians), admixture mapping or mapping by admixture linkage disequilibrium (MALD) has begun to be demonstrated as a powerful tool for identifying disease-causing genetic variants in recently admixed populations, such as African Americans that have both West African and European American ancestry

(*McKeigue, 2005*). For example, recent admixture mapping studies have identified loci associated with disease risks such as prostate cancer (*Ricks-Santi et al., 2012*), lung cancer (*Schwartz et al., 2011*), and traits like blood pressure/obesity (*Shetty et al., 2012*) in African Americans. Admixture mapping assumes that near a disease causing genetic variant there will be enhanced ancestry from the population that has greater risk of getting the disease (*Patterson et al., 2004*). Therefore, by calculating the proportion of ancestry along the genome, one could use that information to identify disease causing loci in an admixed population with low resolution. Subsequent fine mapping restricted to the identified genomic regions may greatly increase the power of the study.

It has been demonstrated that 1,500–2,500 ancestry-informative markers (AIMs) with genome-wide coverage would be sufficient (*Winkler, Nelson & Smith, 2010*) to identify the ancestral chromosome segments for recently admixed populations. To leverage on the power of admixture mapping in African American for identifying disease causing genetic variants that may explain health disparities between populations, panels of AIMs have been proposed for commercially-available high throughput genotyping arrays including the Affymetrix SNP 6.0 and Illumina 1M (*Chen et al., 2010*; *Tandon, Patterson & Reich, 2011*). These genotyping arrays however are likely biased to genetic variations detected from Caucasian samples. The Affymetrix Pan-African array, which interrogates approximately 2.3 million SNPs, was designed for a much greater coverage of genetic variations in African individuals. A panel of AIMs based on the Pan-African array may enhance the distinguishing of parental populations as well as improve genome coverage. Recent advances in statistical genetics have begun to allow admixture mapping utilizing the entire panel of genotyped SNPs (*Baran et al., 2012*; *Churchhouse & Marchini, 2013*; *Maples et al., 2013*), however, we reasoned that providing a panel of AIMs may particularly benefit studies of a limited sample size, help identify unsuitable individuals by genotyping the most informative markers before starting a large GWAS, or guide larger scale targeted re-sequencing projects to pinpoint causal variants. We describe here AIMs identified for the Affymetrix Pan-African array based on Shannon Information Content (SIC) or $F_{st}$ using the 1000 Genomes Project (*Abecasis et al., 2010*) data as references for parental populations.

## MATERIALS AND METHODS

### SNPs covered on the Pan-African array

The Affymetrix Axiom Genome-Wide Pan AFR Genotyping platform (Pan-African array) (Affymetrix, Inc., Santa Clara, California) covers ∼2.3 million SNPs optimized for individuals of African ancestry. The Pan-African array was designed to offer ≥90% coverage of SNPs on the Yoruba genome with minor allele frequency (MAF) greater than 2%. Annotations for the Pan-African array can be accessed at the Affymetrix website (http://www.affymetrix.com/). As a platform optimized for individuals of African individuals, the Pan-African array has been extensively validated in African populations from the HapMap Project (*Altshuler et al., 2010*), including the Luhya from western Kenya (LWK), Maasai from eastern Kenya (MWK), Yoruba from Ibadan, Nigeria (YRI), and the African Ancestry in the Southwest USA (ASW) (*Lu et al., 2011*). This platform offers high genomic

coverage (>85%) in admixed populations with West African ancestry, thus particularly suitable for genome-wide scans in African American individuals (admixture of African and European populations).

## Obtaining allele frequency and genetic map distances on parental populations

Genotypes for 2,176,716 SNPs covered by the Pan-African array were extracted from the 1000 Genomes Project (*Abecasis et al., 2010*) Phase I data for the 85 CEU (Caucasian residents from Utah, USA) and 88 YRI unrelated samples, representing the two major parental populations for African Americans (Western Africans and Europeans). Genome-wide genetic map distances of SNPs for genome assembly GRCh37 (*Frazer et al., 2007*) were downloaded from the website (http://bochet.gcc.biostat.washington.edu/beagle/genetic_maps).

## Selection of ancestry-informative markers

We aimed to pick the SNPs that were expected to provide the highest mutual information content to ancestry or fixation index (i.e., $F_{st}$, a measure of population differentiation due to genetic structure) in the genome using an iterative procedure, conditional on the observed allele frequencies in the 1000 Genomes Project CEU and YRI samples.

(a) *Calculation of mutual information content*: Allele frequencies for the CEU and YRI samples were used to calculate the Shannon Information Content (SIC) for each SNP using a formula from previous studies (*Smith et al., 2004*; *Tandon, Patterson & Reich, 2011*),

$$SIC = -\sum_{i=0}^{1}(a_{i0} + a_{i1})\ln(a_{i0} + a_{i1}) - \sum_{j=0}^{1}(a_{0j} + a_{1j})\ln(a_{0j} + a_{1j}) + \sum_{i=0}^{1}\sum_{j=0}^{1}a_{ij}\ln(a_{ij}),$$

where $a_{00} = (1-m) \times p^{YRI}$, $a_{01} = (m \times p^{CEU})$, $a_{10} = (1-m) \times (1-p^{YRI})$, and $a_{11} = m \times (1-p^{CEU})$. Here, $p^{CEU}$ and $p^{YRI}$ are the allele frequencies in the CEU (European) and YRI (African) samples, and $m$ is the proportion of European ancestry in African Americans, which was set to 0.20 following the same assumption of 20% European ancestry (*Tandon, Patterson & Reich, 2011*). Notably, SNP selection was found not very sensitive to the choice of $m$ (*Smith et al., 2004*). In addition, the $F_{st}$ was also computed for each of the 2,176,716 SNPs between the two parental populations based on Wright's approximate formula (*Wright, 1950*),

$$F_{ST} = (H_T - H_S)/H_T,$$

where $H_T$ represents expected heterozygosity per locus of the total population and $H_S$ represents expected heterozygosity of a subpopulation.

(b) *Selection of AIMs*: We aimed to detect AIMs that are not packed around certain genomic regions due to linkage disequilibrium (LD), thus being more representative of the genome. Since LD declines gradually with increased genetic distance (*Shifman et al., 2003*), we assume each SNP is not in LD with distant SNPs more than 0.25 cm (~250 kb) away, similar to what was used in previous publications (*Tandon, Patterson & Reich, 2011*). We selected AIMs using an iterative procedure for each chromosome: (1) SNPs were ranked

based on SIC; (2) SNP with the highest SIC was selected as a candidate AIM; (3) Any SNPs within 0.25 cm or within 250 kb of the selected SNP were excluded; (4) Steps 2 and 3 were repeated until no more SNPs left. To avoid densely packed markers, no more than 8 candidate AIMs were selected within any 4 cm region. This procedure ensured a good coverage of AIMs across the entire genome. The quality of the detected candidate AIMs was examined using the build-in data quality checking procedure of ANCESTRYMAP 2.0 (*Patterson et al., 2004*) for extracting top "bad" markers, for which allele counts for the ancestral (African and European) genotypes appeared to be grossly inconsistent with counts on the 56 unrelated ASW samples from 1000 Genome Project (*Abecasis et al., 2010*). After applying the ANCESTRYMAP quality checks, we obtained the final panel of AIMs. We also repeated the same selection procedure using $F_{st}$ to identify a companion panel of AIMs. Tables S1 and S2 contain detailed information on the final AIMs.

## Evaluation of the detected AIMs for the Pan-African array

The informativeness of the AIMs was evaluated at each SNP using the ANCESTRYMAP-generated *rpower* value, which is a measure of uncertainty in ancestry inference at a given locus. Specifically, *rpower* is the expected value of the squared correlation between inferred and true ancestry (*Patterson et al., 2004*). In addition, proportion of variance explained (PVE) by the first principal component (PC) using the detected AIMs on the CEU, YRI, and ASW samples was compared with PVEs from previously published AIMs (based on Affymetrix SNP 6.0 and Illumina 1M arrays) (*Tandon, Patterson & Reich, 2011*) as well as 1000 random sets of SNPs.

## RESULTS AND DISCUSSION

Given that the Pan-African array was population-optimized, this platform is expected to offer higher coverage of genetic variation for individuals of African ancestry than previous platforms mostly designed based on Caucasians. Genotyping using the Affymetrix Pan-African array will provide opportunities for performing admixture mapping in African Americans to detect genetic variants associated with those traits that exhibit disparities between parental populations, for instance certain cancers (*Schwartz et al., 2011*). The primary result from this study was a panel of SNPs based on the Pan-African array. We acknowledge that with recent advances in statistical genetics, admixture mapping in African Americans may not rely on a limited number of AIMs any more (*Baran et al., 2012*; *Churchhouse & Marchini, 2013*; *Maples et al., 2013*). We propose that some applications for our detected AIMs could include: (1) to facilitate admixture mapping in limited samples; (2) to help identify problematic individuals through genotyping some top-ranking AIMs before starting a large GWAS; (3) to guide targeted re-sequencing projects that may not have genome-wide genotypic data.

Using an iterative selection algorithm, a total of 6,011 candidate AIMs were detected based on SIC, which can measure the uncertainty in genome-wide ancestry or ancestry at a given locus (*Tandon, Patterson & Reich, 2011*). We further examined the quality of these candidates using the build-in checking procedure of ANCESTRYMAP (*Patterson et al., 2004*) and identified a final set of AIMs with 5995 SNPs based on SIC. We also repeated the

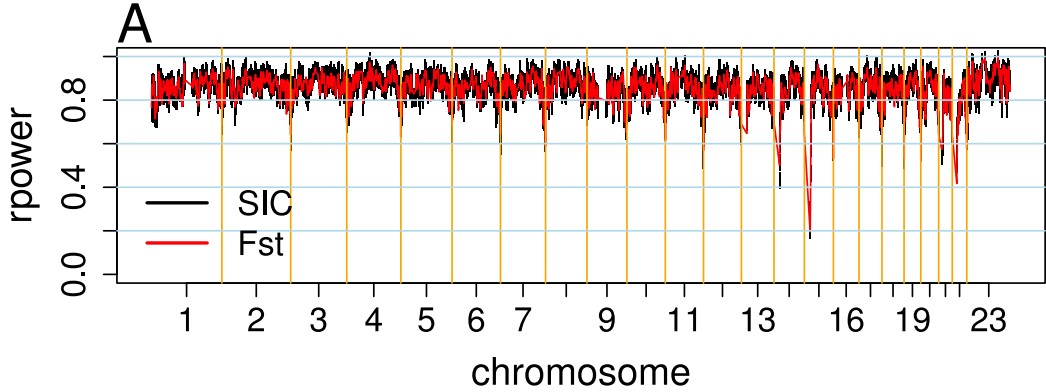

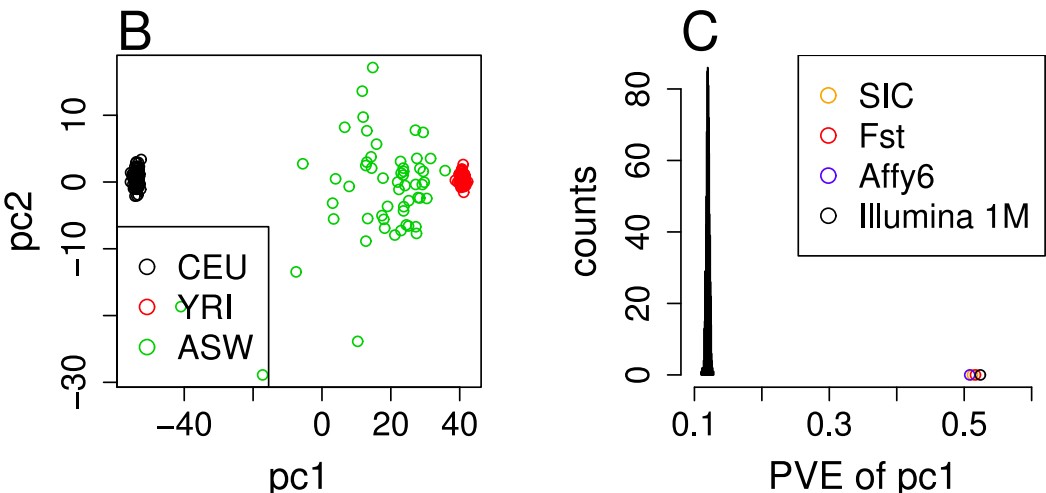

**Figure 1 Evaluation analysis of ancestry-informative markers.** (A) The *rpower* distributions for AIMs selected based on SIC and $F_{st}$. The average *rpower* is 0.85 (*sd* = 0.06) for both lists. (B) Principal components analysis on the 1000 Genomes Project CEU, YRI and ASW panels (*n* = 85 88, 56 unrelated samples, respectively) using the AIMs detected based on SIC. (C) Comparison of the proportion of variance explained (PVE) by the first PCs derived from the CEU, YRI, and ASW samples. The histogram shows the distribution from 1000 randomly-sampled sets of SNPs according to the number of AIMs (based on SIC) on each chromosome. Circles denote real PVE observations for each panel of AIMs: AIMs selected by SIC (5885 SNPs) and $F_{st}$ (6012 SNPs) from Pan-African array, AIMs selected from Affymetrix SNP 6.0 (4290 SNPs), and Illumina 1M (4285 SNPs), respectively.

same analysis using $F_{st}$ to identify a companion panel of 6012 after ANCESTRYMAP checking from 6034 detected candidate SNPs. The selected AIMs with rs numbers, genomic positions, reference alleles, alternative alleles, and allele counts in the CEU or YRI samples are shown in Tables S1 and S2. Overall, AIMs based on SIC and $F_{st}$ performed consistently with each other. The average *rpower* (i.e., average ancestry information) of the AIMs based on SIC or $F_{st}$ was 0.85 (Fig. 1A), compared to ∼0.81 for previous AIMs detected for Affymetrix SNP 6.0 and Illumina 1M arrays (*Tandon, Patterson & Reich, 2011*). The average proportion of European ancestry in ASW was estimated to be 0.25 and 0.24 and the average generations of admixture was estimated to be 5.4 and 5.5 using the AIMs

based on SIC and $F_{st}$, respectively, consistent with previous estimation (*Tandon, Patterson & Reich, 2011*).

The availability of dense genetic variation data from the HapMap Project (*HapMap, 2003*; *HapMap, 2005*) allows a genome-wide analysis of population differentiation. In particular, the CEU (European) and YRI (African) samples represented the two major parental populations of African Americans. Our major criteria of identifying AIMs were designed (1) to enrich SNPs with higher information content (or $F_{st}$) between the CEU and YRI samples; and (2) to have a comprehensive genomic coverage. The genome-wide iterative scan for AIMs based on a genetic distance bin in a size of 0.25 cm, guaranteed a comprehensive coverage of the entire human genome, as well as limit the possibility that the identified AIMs are in strong LD in a particular genomic region, as described in previous publications (*Chen et al., 2010*; *Tandon, Patterson & Reich, 2011*). The final AIMs are those SNPs with the highest SIC (or $F_{st}$) separated by at least the distance of 0.25 cm (~250 kb) between the two parental populations. The detected AIMs were able to recapture the most prominent population structures by being tested on the combined HapMap CEU, YRI, and ASW samples (Fig. 1B). A simulation analysis demonstrated that the detected AIMs based on the Pan-African array explained substantially higher proportion of variance by the first PC across the same populations than random sets of SNPs from the array (Fig. 1C). Though our analysis showed that the AIMs detected based on SIC and $F_{st}$ performed consistently, given some potential problems of $F_{st}$, in particular its dependency on within-population diversity (*Sherwin, 2010*), we generally recommend the use of the final panel of AIMs detected based on SIC.

The assumption of no LD based on 0.25 cm (~250 kb) could be stringent and cause loss of some informative SNPs, given that the average distance of LD decay between SNP pairs is around 20–30 kb across diverse populations, with generally shorter distance in African Americans (*Shifman et al., 2003*). Nevertheless, this cutoff was chosen to balance between minimizing the possibility of LD and the comprehensive genomic coverage of AIMs (*Tandon, Patterson & Reich, 2011*).

In summary, the Affymetrix Pan-African array provides a population-optimized genotyping platform for GWAS in individuals of African ancestry. The genotypic data profiled by this platform also offers opportunities for admixture mapping in African Americans, a recently admixed population, for certain complex traits and disease susceptibilities with disparities between parental populations. The AIMs we described in this study represent the most informative sets of unlinked markers that can be an important resource to facilitate such applications based on this new tool.

### Funding

This work is partially supported by a grant R21HG006367 (to WZ) from the National Institutes of Health. The funders had no role in study design, data collection and analysis, decision to publish, or preparation of the manuscript.

## Grant Disclosures

The following grant information was disclosed by the authors:

National Institutes of Health: R21HG006367.

## Competing Interests

The authors declare there are no competing interests.

## Author Contributions

- Xu Zhang performed the experiments, analyzed the data, contributed reagents/materials/analysis tools, wrote the paper, prepared figures and/or tables, reviewed drafts of the paper.
- Wenbo Mu and Cong Liu performed the experiments, analyzed the data.
- Wei Zhang conceived and designed the experiments, performed the experiments, analyzed the data, contributed reagents/materials/analysis tools, wrote the paper, prepared figures and/or tables, reviewed drafts of the paper.

## Supplemental Information

Supplemental information for this article can be found online at http://dx.doi.org/10.7717/peerj.660#supplemental-information.

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
