# Peer review of "Ancestry-informative markers for African Americans based on the Affymetrix Pan-African genotyping array"

_PeerJ, doi:10.7717/peerj.660_

## Round 0.1 · original submission · Major Revisions

I see the strength of this study as it presents a list of ~2,500 AIMs for African Americans based on the Affymetrix Genome-Wide PanAFR Array. However, I recommend major revisions of this manuscript to improve its quality.

·

Basic reporting

The paper is well written, concise, and clear. The authors have included sufficient background of the study, which is straightforward enough to a broad audience.
One minor issue: though this might be well known to the field, the assumption of admixture mapping (Page 4, Lines 84-86) might need a proper reference (PMC1181990).

Experimental design

The authors have provided a detailed and thorough descriptions of the database used and pipeline for selection of AIMs, which can be easily reproduced by other investigators.

Validity of the findings

The authors have illustrated the comprehensive coverage of the proposed AIMs and the successful performances in differentiating the mixed populations. If possible, it would be more persuasive to briefly present and compare what previous related study (mentioned in the introduction section) have achieved so as to further demonstrate the usefulness of the proposed AIMs.

Additional comments

This study offers an updated panel of AIMs with genome-wide coverage, based on an African population optimized genotyping platform. The 2,532 AIMs identified in the study will serve as a powerful database to help track disease risk alleles in African American populations, as well as provide clues that may help elucidate the mechanisms of various diseases. I therefore recommend accepting with the above minor revisions.

·

Basic reporting

Line 87: Local ancestry can be used to infer a disease causing locus, but is ill suited for identifying specific variants.

Line 119: "cent Morgan" should read "Centimorgan"

Line 160: You mention sliding windows here, but the Methods section, line 132, makes it seem like the bins were fixed, rather than sliding.

Line 169: 20-30 kb seems like a very small window. Shifman does say, "At larger distances (40–80 kb), only ∼14% of the pairs showed no evidence of recombination." This however leaves a lot of room for significant LD. "No evidence of recombination" is equivalent to saying LD = 1.

Figure 2 does not provide a convincing measure of this panel's validity and could be removed (see comments in Validity of Findings section, Lines 165-166). If a similar analysis were done with a random selection of markers from the Pan-African array and shown to infer global ancestry less well, then the two PCA plots placed side-by-side might be a good replacement.

Experimental design

Line 47: Actually, admixture mapping no longer requires a sparse marker panel of AIMs. There are a number of options available for dense marker panels, that can handle markers which are linked in the ancestral populations.

Line 96: This is true for GWAS, but I doubt it would add much to the power of a MALD study. The ability to infer local ancestry in African Americans is quite good, even with European-centric marker panels. Since the association sought is between phenotype and ancestry, rather than between phenotype and a particular variant, untagged genetic variation can still be mapped using MALD.

Line 128: This is less true than it was a few years ago. When local LD is properly modeled, strongly linked marker data can be used instead of AIMs panels.

Validity of the findings

Lines 145 -147: As mentioned in the Experimental Design comments, untagged genetic variation can still be mapped using MALD, so the increased genetic variation captured in the Pan-African array does not necessarily make this panel better than others already available.

Lines 165-166: I'm not sure this tells us much about the validity of this panel. Global ancestry can be quite accurately inferred with randomly selected markers.

Most admixture software uses genetic position in cM. Adding this information to Supplementary Table S1 would be useful.

Additional comments

While the methods used to generate this panel appear to be sound, claims about the necessity of an AIMs panel for MALD need to be addressed. There are a number of algorithms and software packages available that would support admixture mapping of the entire Pan-African array, without needing to thin the data. I would expect that using all markers in the Pan-Afrian array would be more powerful than limiting the analysis to a small fraction of the available markers. The merits of using the sparse marker panel presented here need to be addressed before recommending acceptance.

Reviewer 3 ·

Basic reporting

No Comments.

Experimental design

No Comments.

Validity of the findings

No Comments.

Additional comments

African populations are genetically more diverse than European and Asian populations. GWAS in African American populations pose unique challenges given the extensive genetic heterogeneity. Ancestry-informative markers (AIMs) for African Americans will help to estimate ancestry proportions in this population. In this study, the authors proposed a list of ~2,500 AIMs for African Americans based on the Affymetrix Genome-Wide PanAFR Array. Affymetrix Genome-Wide PanAFR Array was designed to maximize coverage of novel common and rare variants in populations of Yoruba ancestry. This array is also capable of determining the alleles of other people from West Africa, East Africa and even African Americans. So AIM list based on PanAFR Array will benefit the genetic researchers who is using this array. Even though this article is pretty condensed, the manuscript was well-written. The methods and conclusions are sound. I don't have any major concerns. I only have two minor comments.

1. It's better for the authors to describe the formula of FST, which may help the readers who are not familiar with this term.

2. In Figure 2, I suggest the authors to provide the eigenvalues (or percentages of variance) for each component.

---

## Round 0.2 · Major Revisions

After receiving remarks from reviewers, I recommend you to arrange changes and explanations on these remarks.

·

Basic reporting

The background provided is clear and well written. More recent work has been done, however, that the authors have not included. Lines 140-144, for example, discuss limitations on admixture mapping that no longer apply when using the appropriate software. The following describe a few methods and software packages that can analyze the entire Pan-African array without needing to thin the data:

Price, A. L., Tandon, A., Patterson, N., Barnes, K. C., Rafaels, N., Ruczinski, I., et al. (2009). Sensitive detection of chromosomal segments of distinct ancestry in admixed populations. PLoS Genetics, 5(6), e1000519. doi:10.1371/journal.pgen.1000519.

Baran, Y., Pasaniuc, B., Sankararaman, S., Torgerson, D. G., Gignoux, C., Eng, C., et al. (2012). Fast and accurate inference of local ancestry in Latino populations. Bioinformatics, 28(10), 1359–1367. doi:10.1093/bioinformatics/bts144

Maples, B. K., Gravel, S., Kenny, E. E., & Bustamante, C. D. (2013). RFMix: A Discriminative Modeling Approach for Rapid and Robust Local-Ancestry Inference. The American Journal of Human Genetics. doi:10.1016/j.ajhg.2013.06.020

Churchhouse, C., & Marchini, J. (2013). Multiway admixture deconvolution using phased or unphased ancestral panels. Genetic Epidemiology, 37(1), 1–12. doi:10.1002/gepi.21692

Experimental design

No comments

Validity of the findings

The AIMs panel and the methods used to identify the AIMs are valid, but limiting an analysis to these AIMs would result in a decrease in statistical power when compared to using the entire Pan-African array.

Additional comments

While the methods used to generate this panel appear to be sound, claims about the necessity of an AIMs panel for MALD need to be addressed. There are a number of algorithms and software packages available that would support admixture mapping of the entire Pan-African array, without needing to thin the data (see some examples above). I would expect that using all markers in the Pan-Afrian array would be more powerful than limiting the analysis to a small fraction of the available markers. The merits of using the sparse marker panel presented here need to be addressed in contrast to using the entire Pan-African array before recommending acceptance.

Reviewer 3 ·

Basic reporting

No Comments

Experimental design

No Comments

Validity of the findings

No Comments

Additional comments

Generaly, I am all right with the revised version. The manuscript was well-written. Also, the authors included the detailed method for FST as I suggested for the first round.

---

## Round 0.3 · accepted · Accept

After carefully, studying your comments and changes, I have come up to conclusion that your MS is accepted now. Congratulations.